# New Series of Double-Modified Colchicine Derivatives: Synthesis, Cytotoxic Effect and Molecular Docking

**DOI:** 10.3390/molecules25153540

**Published:** 2020-08-02

**Authors:** Julia Krzywik, Maral Aminpour, Ewa Maj, Witold Mozga, Joanna Wietrzyk, Jack A. Tuszyński, Adam Huczyński

**Affiliations:** 1Department of Medical Chemistry, Faculty of Chemistry, Adam Mickiewicz University, Uniwersytetu Poznańskiego 8, 61-614 Poznań, Poland; julia.krzywik@amu.edu.pl; 2TriMen Chemicals, Piłsudskiego 141, 92-318 Łódź, Poland; mozga@trimen.pl; 3Department of Oncology, University of Alberta, Edmonton, AB T6G 1Z2, Canada; aminpour@ualberta.ca (M.A.); jack.tuszynski@gmail.com (J.A.T.); 4Hirszfeld Institute of Immunology and Experimental Therapy, Polish Academy of Sciences, Rudolfa Weigla 12, 53-114 Wrocław, Poland; ewa.maj@hirszfeld.pl (E.M.); joanna.wietrzyk@hirszfeld.pl (J.W.); 5DIMEAS, Politecnico di Torino, Corso Duca degli Abruzzi, 24, 10129 Torino, Italy

**Keywords:** anticancer agents, colchicine derivatives, reductive alkylation, tubulin inhibitors, docking studies

## Abstract

Colchicine is a well-known anticancer compound showing antimitotic effect on cells. Its high cytotoxic activity against different cancer cell lines has been demonstrated many times. In this paper we report the syntheses and spectroscopic analyses of novel colchicine derivatives obtained by structural modifications at C7 (carbon-nitrogen single bond) and C10 (methylamino group) positions. All the obtained compounds have been tested in vitro to determine their cytotoxicity toward A549, MCF-7, LoVo, LoVo/DX, and BALB/3T3 cell lines. The majority of obtained derivatives exhibited higher cytotoxicity than colchicine, doxorubicin and cisplatin against the tested cancerous cell lines. Additionally, most of the presented derivatives were able to overcome the resistance of LoVo/DX cells. Additionally, their mode of binding to β-tubulin was evaluated *in silico.* Molecular docking studies showed that apart from the initial amides **1** and **2**, compound **14**, which had the best antiproliferative activity (IC_50_ = 0.1–1.6 nM), stood out also in terms of its predicted binding energy and probably binds best into the active site of βI-tubulin isotype.

## 1. Introduction

Tubulin is the target of some of the most widely used anticancer anti-mitotic agents, such as vinblastine or paclitaxel [1,2]. Unfortunately, clinical usefulness of many microtubule disrupting agents has been reduced as a result of tumor cell multidrug-resistance (MDR). The main role in MDR is played by P-glycoprotein, which is overexpressed in various types of cancers and is known to be associated with poor response to chemotherapy [3,4,5]. Colchicine **1** (see Scheme 1) is a plant-based alkaloid extracted from *Colchicum autumnale* and *Gloriosa superba* that shows antimitotic effects on a number of cancer cell lines. It binds to tubulin, inhibiting its assembly and microtubule polymerization and finally arresting cell division at metaphase [6,7,8,9,10,11,12]. However, in addition to the problem with cancer cells developing drug resistance, colchicine was found to have some toxic effects, including accumulation in the gastrointestinal tract, as well as neurotoxicity [6,13,14,15,16,17,18,19]. These are the two main disadvantages that limit the use of colchicine in chemotherapy.

Despite the above limitations, colchicine exhibits high antiproliferative activity and represents an interesting research area for understanding of relationship between the structure and biological activity (SAR) of compounds interacting with tubulin. Many colchicine analogues containing amide or urethane at position C7 have been studied [20,21,22,23,24,25,26,27]. Such groups may be hydrolyzed in vivo causing a change in pharmacological activity [28,29]. Conversion of such moieties into single carbon-nitrogen bond prevents hydrolysis of the side chain in the cells and creates a new class of colchicine derivatives. Numerous modified colchicine and thiocolchicine derivatives with substituted benzyl moieties at carbon C7 have been synthesized and their biological activity has been evaluated. Some of the described derivatives showed antiproliferative activity superior to that of colchicine [22,30]. In addition, as shown in our earlier studies, the replacement of -OCH_3_ group at position C10 with -NHCH_3_ group leads to generation of compounds more active than unmodified colchicine [20]. On the basis of these findings, we decided to develop the concept, design, and synthesize a new series of double-modified colchicine derivatives with different amine moieties at C7 atom (aromatic but also alkyl substituents that have not been previously widely studied) and with a methylamino group at C10 atom in the tropolone ring C.

Herein, we report the syntheses and spectroscopic analyses of a series of structurally different derivatives of colchicine obtained by its modification at C7 (amide bond replacement by a carbon-nitrogen single bond) and C10 (methylamino group) positions. We also describe the results of in vitro antiproliferative activity evaluation of the starting compounds **1**–**2** and the obtained colchicine derivatives (**3**–**17**) against four human cancer cell lines, namely A549, MCF-7, LoVo, and LoVo/DX as well as normal murine cells BALB/3T3. To gain more knowledge about molecular mechanism of action of the investigated compounds (**1**–**17**), we also present *in silico* results of a molecular docking study into the colchicine-binding site (CBS) of β-tubulin. The derivatives presented in this work are double-modified colchicine derivatives with the amine groups at C7 and C10 positions. On the basis of the obtained preliminary structure–activity relationship (SAR) studies, we will be able to design new promising compounds for more extended research.

## 2. Results and Discussion

### 2.1. Chemistry

To determine the effect of replacement of the amide moiety at position C7 by a single nitrogen-carbon bond and, at the same time, -OCH_3_ group replacement by -NHCH_3_ at position C10 of colchicine **1**, on its bioactivity, fourteen new derivatives (**4**–**17**) were designed. The obtained compounds contained different moieties (replacing acetamide of unmodified compound **1**) attached to nitrogen at position C7, such as: alkyl chains of various length, straight or branched (**4**–**8**), alkyl chains containing various substituents (**9**–**12**), aromatic moiety with or without substituents (**13**–**15**) or with heteroatom in the ring (**16**–**17**). Diversity of groups at position C7 is intended to facilitate the analysis of the structure–activity relationship. The synthetic routes to colchicine derivatives **2**–**17** are outlined in Scheme 1.

Compound **3** was readily available from **1** by treatment with methylamine and followed by hydrolysis with 2M HCl [20,31]. *N*-deacetyl-10-methylamino-10-demethoxycolchicine **3** became the starting material for the synthesis of new derivatives (**4**–**17**). On the basis of the method described previously [22], reductive alkylation of **3** in a reaction with the corresponding aldehyde in the presence of sodium cyanoborohydride allowed us to obtain double-modified compounds **4**–**9** and **11**–**17**. However, aldehydes used in the present paper did not require activation with acid, so the reactions were carried out without the addition of acetic acid, which eliminated the formation of unidentified by-products. Most of the aldehydes necessary for the synthesis of the designed compounds were purchased, but aldehydes for the synthesis of compounds **4** and **11** were obtained by oxidation of the corresponding alcohols using pyridinium chlorochromate, PCC [32]. Compound **10** was synthesized via addition of amine (**3**) to the Michael acceptors (acrylonitrile). All obtained compounds (**2**–**17**) were isolated in pure form after column flash chromatography on silica gel.

The structures and purities of all products **2**–**17** were determined using the LC-MS, ^1^H and ^13^C NMR analyses, and are shown in Appendix A and discussed below. The signals corresponding to -OCH_3_ group at position C10 of **1** in the ^1^H NMR and ^13^C NMR spectra were observed as singlets at 4.0 ppm and at 56.5 ppm, respectively. After replacement of this group in the tropolone ring of colchicine by -NHCH_3_ group in compounds **2**–**17**, new signals appeared, at 3.0–3.1 ppm as a doublet and 7.2–7.3 ppm as a quartet in ^1^H NMR spectra and at 29.5–29.9 ppm in ^13^C NMR spectra. The signals characteristic of the acetyl group in colchicine **1** appeared at 1.9 ppm (-CH_3_) and 8.6 ppm (-NH-) in ^1^H NMR and at 22.7 ppm (-CH_3_) and 170.3 ppm (-C=O) in ^13^C NMR, respectively. These signals are no longer observed in the NMR spectra of **3**–**17** derivatives. The signal assigned to the hydrogen atom at C7 has been shifted from 4.6 ppm for unmodified colchicine **1** to 3.4–3.6 ppm for compounds **3**–**17**. The ESI mass spectrometry confirmed the structure of the obtained analogs by the presence of *m/z* signals assigned to the corresponding pseudomolecular ions of these compounds.

### 2.2. In Vitro Determination of Drug-Induced Inhibition of Human Cancer Cell Line Growth

The tested compounds (**1**–**17**) were evaluated for their in vitro antiproliferative effect on four human cancer cell lines and one normal murine embryonic fibroblast cell line according to the previously published procedure [20]. Detailed information concerning antiproliferation assay can be found in Appendix A.

All obtained derivatives showed better antiproliferative activity than those of the commonly used antitumor agents doxorubicin and cisplatin. It is also worth noting that all compounds (except **7** with long alkyl chain) were characterized by very high cytotoxicity with IC_50_ ≤ 10 nM towards three of the four tumor lines tested (see Table 1).

Even though compound **7** has a weak potential compared to the other derivatives with an amine moiety on C7 carbon and methylamino group on C10 carbon, its IC_50_ values were in the nanomolar range. The activities of the remaining derivatives against A549 cells were better than that of colchicine **1** and for the majority of compounds comparable with the second starting compound 10-methylaminocolchicine **2**. Compounds **13** (IC_50_ = 4.6 nM) and **14** (IC_50_ = 1.1 nM) were characterized by the lowest IC_50_ values towards the A549 line. In relation to the MCF-7 cell line, the newly designed derivatives (**4**–**17**) showed activities similar to that of colchicine **1** and only one compound—a derivative with a 2-chlorobenzyl moiety **14**—stood out from them having a lower IC_50_ = 0.7 nM, comparable with that of amide **2** (IC_50_ = 1.6 nM). In relation to the LoVo cells, most of the new derivatives exhibited higher activities than unmodified colchicine **1**, and five of them had IC_50_ comparable to amide **2**. Derivative **14** turned out to be the most cytotoxic towards the LoVo cell line (IC_50_ = 0.1 nM). As many as five of the obtained derivatives (**8**, **12**–**14** and **17**) exhibited IC_50_ ≤ 10 nM towards LoVo/DX cells. Moreover, compound **14** (IC_50_ = 1.6 nM) was observed to be the most active towards the doxorubicin-resistant subline LoVo/DX, approx. 440 times more potent than unmodified colchicine **1** (IC_50_ = 702.2 nM). Comparison of the results for C7-amine double-modified analogues and those of previously published studies of C7-amide double-modified analogues [20] showed that in both cases some of the most cytotoxic compounds were derivatives containing a 2-chlorobenzyl ring at position C7. These results are noteworthy and suggest that extended and more detailed research is necessary to determine the meaning and role of the above mentioned substituent. Additionally, it should be mentioned that the antiproliferative activity depends on the type of the tested cell lines and is different for different cells. In vitro tests allow the initial determination of biological activity, but in vivo studies are necessary to determine the therapeutic potential.

Selectivity index (SI) is a major challenge in drug discovery, because it defines the ability of a particular compound to preferentially kill tumor cells in relation to normal cells. Therefore, the obtained compounds were tested not only against cancerous cells but also against the normal murine embryonic fibroblast cell line (BALB/3T3). The majority of the tested derivatives were characterized with low SI values (see Figure 1), smaller than those found for the starting compounds **1**–**2**. However, derivative **14** (with 2-chlorobenzyl moiety at C7 carbon), which stands out from the obtained compounds is noteworthy. Its SI values were especially high for all tested cancer cell lines (SI = 4.1 for A549 cells, SI = 6.1 for MCF-7 cells, SI = 93.7 for LoVo cells, SI = 2.8 for LoVo/DX cells) and were higher than SI of parental amides **1** and **2**. *N*-(2-chlorobenzyl)-10-methylaminocolchicine **14** therefore emerges as a very promising molecule (the lowest IC_50_ and the highest SI) for further more detail research (in vitro but also in vivo) as a potential anticancer agent. Moreover, compounds **10**–**13** and **17** exhibited SI > 3 for LoVo cells. The results indicated that conversion of acetyl at C7 atom of the starting compounds **1** and **2** into a single carbon-nitrogen bond did not bring (except the derivatives mentioned above, in particular **14**) more selective compounds in relation to the cancer cells tested.

The data presented in Table 1 show that the studied compounds inhibited the proliferation of the doxorubicin-resistant subline LoVo/DX less effectively than that of the sensitive LoVo cell line. However, all new derivatives **3**–**17** exhibited lower RI than the starting compounds **1** and **2** with an acetyl at position C7. In addition, as many as eleven of the tested colchicine derivatives (**3**–**8**, **11**–**13**, **16**–**17**) showed RI ≤ 10 (see Figure 2). The best values of the calculated resistance index, RI = 1.1–2.1 were obtained for derivatives **6**–**8** (*N*-butyl/*N*-decyl/*N*-isobutyl 10-methylaminocolchicine), which means that the cells are very sensitive to these compounds. The results indicate that some of the double-modified colchicine derivatives are able to overcome the drug resistance of the LoVo/DX cell line, but this desirable property was offset by poor SI values (see Figure 1). It is possible that a replacement of the amide moiety at position C7 by a single nitrogen-carbon bond will solve one of the main problems of using colchicine in chemotherapy (multidrug resistance) and therefore, such derivatives should be taken into account for further investigation aimed at structure optimization.

### 2.3. In Silico Determination of the Molecular Mode of Action

In our study, computational methodology (according to the previously published procedure [20]) has been employed to precisely explore the molecular basis for the binding of the fourteen novel colchicines to β-tubulin. The β-tubulin monomer together with the highly homologous β-tubulin monomer form a stable dimer, which is the building block of microtubules in the cytoskeleton structure of every eukaryotic cell. Tubulin, in particular β-tubulin, is the target of many antiproliferative agents. In our simulations, compounds described above were docked into the most abundant isotype of β-tubulin in most cancer cells (βI-tubulin) in the colchicine-binding site and ranked according to their predicted binding affinity.

According to our computational predictions, the compounds from the lowest binding energy to the highest are ordered as follows: **1** (−39.1), **2** (−37.5), **14** (−31.5), **16** (−27.4), **15** (−27.1), **11** (−19.9), **6** (−19.4), **4** (−17.2), **7** (−16.9), **5** (−4.9), **3** (−4.5), **17** (−3.6), **9** (−2.6), **8** (−1.0), **10** (5.0), and **12** (8.7) with the binding energies in the parentheses given in kcal/mol units. Binding energies of these compounds are shown in Figure 3 and in Appendix A
Appendix A.

Replacement of the methoxy group in colchicine 1 by the methylamino group in derivative **2** at C10 carbon significantly improved the antiproliferative activity of unmodified **1** (Table 1) but their BE values did not differ significantly (−39.1 kcal/mol for **1** and −37.5 kcal/mol for **2**). Compound **14** was the most toxic to tumor cells (IC_50_ = 0.1–1.6 nM) and has the third lowest energy (−31.5 kcal/mol). These observations may indicate that the new double-modified colchicine **14** fits well to βI-tubulin and may have a similar mechanism of action to **1**, despite significantly lower IC_50_ values. However, if we look at subsequent derivatives with a relatively low BE, **15**–**16** (approx. −27.0 kcal/mol) close to BE for **14** and compare with their IC_50_ values, we notice here the absence of any correlation—compounds **15** and **16** were definitely less active in vitro than compound **14**. In turn, compounds **5**, **8**–**10**, **12** and **17** had the highest BE from among the new derivatives with a single carbon-nitrogen bond at position C7 (**4**–**17**) even though their IC_50_ values were comparable or lower than those of compounds **15** and **16** with low binding energies. Thus, the trend in the computed binding energy values is not the same as for IC_50_ values that were determined from the experimental assays. However, it should be emphasized that IC_50_ is strongly dependent on the ability of these compounds to enter cancer cells through their membranes without being removed by efflux pumps and cause cell death. Since there is little correlation here, we conclude that this ability depends mainly on: solubility and membrane permeability plus affinity to P-glycoproteins (P-gp). It is worth mentioning that colchicine is subject to multidrug resistance in tumor cells as a P-gp substrate. It induces P-gp activity causing a conformational change, which subsequently leads to its rapid efflux from tumor cells [4,33,34]. Binding energy to its primary target, tubulin, is a secondary factor affecting the value of IC_50_ and most of these compounds fall into a fairly narrow range due to a similar mode of binding to their target. On the other hand, SI is the ratio of the IC_50_′s for normal cells versus cancer cells. It largely depends on the three main factors listed above of which solubility and membrane permeability do not change between cancer and normal cells, so it mainly depends on the affinity to P-gp, which is not known to us at present.

Therefore, we can see that the *in silico* results do not correlate with the in vitro determined biological activity of these compounds as indicated by the corresponding IC_50_ values. The inclusion of binding affinity calculations for these compounds with regard to the other tubulin isotypes is still informative in a broader context of tubulin-binding agents and with respect to most important efflux transporters, in order to minimize interactions with the latter and maximize with the former, and hence increase the activity of the chemical compound.

Schematic interactions of the compounds with CBS residues of βI-tubulin are shown in Table 2 and Appendix A
Appendix A. Derivatives **4**–**8** were designed with different alkyl chains at position C7 of 10-methylaminocolchicine **2**. In this group, the common binding residues were Asn100, Leu688, Asn691, and Lys785. Compound **7** showed additionally some interactions with α-tubulin (Gln10, Asn100, Ser177, Ala179). Another series with alkyl chains having various substituents (**9**–**12**) at C7 carbon were bound mostly through Asn100, Thr178, Leu681, Leu688, and Lys785. Compounds **13**–**17** have been designed with an aromatic group side chain without or with substituents at C7 position. The binding residues in this group belong to both α-tubulin (Asn100, Thr178) and β-tubulin (Leu681, Asn682, Asp684, Leu688, Lys785).

## 3. Materials and Methods

### 3.1. General

Information concerning reagents and solvents can be found in the Appendix A.

### 3.2. Spectroscopic Measurements

Information concerning equipment used for measurements can be found in the Appendix A.

### 3.3. Synthesis

#### 3.3.1. Synthesis of **2** and **3**

Synthesis of 10-*N*-methylaminocolchicine **2** and *N*-deacetyl-10-methylamino-10- demethoxycolchicine **3** was performed according to the previously published procedure [20] and can be found in Appendix A.

#### 3.3.2. General Procedure for the Synthesis of Colchicine Derivatives **5**–**9** and **12**–**17**

Compound **5**–**9** and **12**–**17** were obtained directly from compound **3**. To a solution of compound **3** (1.0 equiv.) in MeOH, the corresponding aldehyde (1.0 equiv.) was added followed in 15 min by NaBH_3_CN (1.2 equiv.). Reaction progress was monitored by LC-MS. Then the mixture was evaporated under reduced pressure, diluted with EtOAc, washed with 5% NaHCO_3_ brine and dried over Na_2_SO_4_. The residue was purified using column flash chromatography (silica gel; EtOAc/MeOH, 20/1 v/v) and next lyophilized from dioxane to give respective compounds.

##### Compound **5**

Yellow solid, yield 47%.

ESI-MS for C_23_H_30_N_2_O_4_ (*m*/*z*): [M + H]^+^ 399, [M + Na]^+^ 421, [2M + H]^+^ 797, [2M + Na]^+^ 819.

^1^H NMR (500 MHz, CDCl_3_) δ 7.64 (s, 1H), 7.36 (d, *J* = 11.0 Hz, 1H), 7.27–7.25 (m, 1H), 6.53–6.48 (m, 2H), 3.91 (s, 3H), 3.88 (s, 3H), 3.55 (s, 3H), 3.41–3.37 (m, 1H), 3.06 (d, *J* = 5.4 Hz, 3H), 2.51–2.47 (m, 1H), 2.45–2.35 (m, 1H), 2.33–2.27 (m, 1H), 2.24–2.15 (m, 2H), 1.72–1.65 (m, 1H), 1.46–1.37 (m, 2H), 0.82 (t, *J* = 7.4 Hz, 3H).

^13^C NMR (126 MHz, CDCl_3_) δ 175.7, 155.1, 152.6, 150.9, 150.6, 141.1, 138.6, 135.5, 130.5, 127.0, 124.4, 107.2, 107.1, 61.3, 61.0, 60.6, 56.1, 49.9, 39.4, 30.5, 29.5, 23.4, 11.8.

##### Compound **6**

Yellow solid, yield 44%.

ESI-MS for C_24_H_32_N_2_O_4_ (*m*/*z*): [M + H]^+^ 413, [M + Na]^+^ 435, [2M + H]^+^ 825, [2M + Na]^+^ 847.

^1^H NMR (500 MHz, CDCl_3_) δ 7.63 (s, 1H), 7.35 (d, *J* = 11.0 Hz, 1H), 7.28–7.26 (s, 1H), 6.50–6.48 (d, *J* = 12.3 Hz, 2H), 3.89 (s, 3H), 3.87 (s, 3H), 3.54 (s, 3H), 3.39–3.37 (m, 1H), 3.05 (d, *J* = 5.3 Hz, 3H), 2.49–2.46 (m, 1H), 2.39–2.37 (m, 1H), 2.30–2.28 (m, 1H), 2.22–2.18 (m, 2H), 1.73–1.64 (m, 1H), 1.39–1.36 (m, 2H), 1.28–1.25 (m, 2H), 0.80 (t, *J* = 7.3 Hz, 3H).

^13^C NMR (126 MHz, CDCl_3_) δ 175.7, 155.1, 152.6, 150.7, 150.5, 141.1, 138.6, 135.5, 130.5, 126.9, 124.3, 107.3, 107.1, 61.3, 61.0, 60.6, 56.0, 47.6, 39.3, 32.3, 30.5, 29.5, 20.4, 14.0.

##### Compound **7**

Yellow solid, yield 40%.

ESI-MS for C_30_H_44_N_2_O_4_ (*m*/*z*): [M + H]^+^ 497, [M + Na]^+^ 519, [2M + H]^+^ 993, [2M + Na]^+^ 1015.

^1^H NMR (500 MHz, CDCl_3_) δ 7.64 (s, 1H), 7.36 (d, *J* = 11.1 Hz, 1H), 7.32–7.27 (m, 1H), 6.52–6.48 (m, 2H), 3.90 (s, 3H), 3.87 (s, 3H), 3.55 (s, 3H), 3.44–3.37 (m, 1H), 3.06 (d, *J* = 5.4 Hz, 3H), 2.55–2.48 (m, 1H), 2.43–2.37 (m, 1H), 2.34–2.26 (m, 1H), 2.25–2.17 (m, 2H), 1.75–1.67 (m, 1H), 1.44–1.36 (m, 2H), 1.26–1.15 (m, 14H), 0.83 (t, *J* = 7.0 Hz, 3H).

^13^C NMR (126 MHz, CDCl_3_) δ 175.6, 155.1, 152.7, 150.5, 141.1, 138.7, 135.4, 130.5, 126.9, 124.3, 107.3, 107.1, 61.3, 61.1, 60.6, 56.0, 48.0, 39.3, 31.9, 30.4, 29.6, 29.5, 29.3, 27.3, 22.7, 14.1.

##### Compound **8**

Yellow solid, yield 45%.

ESI-MS for C_24_H_32_N_2_O_4_ (*m*/*z*): [M + H]^+^ 413, [M + Na]^+^ 435, [2M + H]^+^ 825, [2M + Na]^+^ 847.

^1^H NMR (500 MHz, CDCl_3_) δ 7.68 (s, 1H), 7.36 (d, *J* = 11.0 Hz, 1H), 7.25–7.21 (m, 1H), 6.55–6.47 (m, 2H), 3.92 (s, 3H), 3.89 (s, 3H), 3.56 (s, 3H), 3.38–3.34 (m, 1H), 3.07 (d, *J* = 5.4 Hz, 3H), 2.42–2.38 (m, 1H), 2.33–2.29 (m, 2H), 2.24–2.15 (m, 1H), 2.08–2.04 (m, 1H), 1.73–1.68 (m, 1H), 1.64–1.62 (m, 1H), 0.88 (d, *J* = 6.6 Hz, 3H), 0.82 (d, *J* = 6.6 Hz, 3H).

^13^C NMR (126 MHz, CDCl_3_) δ 175.8, 155.1, 152.6, 151.1, 150.6, 141.1, 138.5, 135.5, 130.5, 127.0, 124.6, 107.2, 107.0, 61.3, 61.2, 60.6, 56.0, 56.0, 39.4, 30.5, 29.5, 28.8, 20.8, 20.7.

##### Compound **9**

Yellow solid, yield 50%.

ESI-MS for C_22_H_28_N_2_O_5_ (*m*/*z*): [M + H]^+^ 401, [M + Na]^+^ 423, [2M + H]^+^ 801, [2M + Na]^+^ 823.

^1^H NMR (500 MHz, CDCl_3_) δ 7.62 (s, 1H), 7.39 (d, *J* = 11.1 Hz, 1H), 7.35–7.33 (m, 1H), 6.57–6.50 (m, 2H), 3.90 (s, 3H), 3.88 (s, 3H), 3.64–3.57 (m, 2H), 3.55 (s, 3H), 3.48–3.44 (m, 1H), 3.13 (s, 1H), 3.07 (d, *J* = 5.4 Hz, 3H), 2.7–2.74 (m, 1H), 2.45–2.40 (m, 2H), 2.35–2.25 (m, 2H), 1.75–1.71 (m, 1H).

^13^C NMR (126 MHz, CDCl_3_) δ 175.4, 155.2, 152.8, 150.5, 150.4, 141.4, 139.1, 135.3, 130.7, 126.7, 124.0, 107.8, 107.2, 61.3, 61.2, 60.8, 60.7, 56.1, 49.6, 39.3, 30.4, 29.5.

##### Compound **12**

Yellow solid, yield 55%.

ESI-MS for C_23_H_27_F_3_N_2_O_4_ (*m*/*z*): [M + H]^+^ 453, [M + Na]^+^ 475, [2M + H]^+^ 905, [2M + Na]^+^ 927.

^1^H NMR (500 MHz, CDCl_3_) δ 7.66 (s, 1H), 7.37 (d, *J* = 11.1 Hz, 1H), 7.29–7.26 (m, 1H), 6.53–6.50 (m, 2H), 3.91 (s, 3H), 3.89 (s, 3H), 3.57 (s, 3H), 3.43–3.38 (m, 1H), 3.07 (d, *J* = 5.4 Hz, 3H), 2.80–2.68 (m, 1H), 2.58–2.50 (m, 1H), 2.43–2.37 (m, 1H), 2.35–2.29 (m, 1H), 2.27–2.14 (m, 3H), 1.73–1.68 (m, 1H).

^13^C NMR (126 MHz, CDCl_3_) δ 175.7, 155.1, 152.8, 150.6, 150.0, 141.2, 138.9, 135.3, 130.3, 127.7, 126.7, 124.0, 107.3, 107.1, 61.3, 60.9, 60.6, 56.0, 40.8, 40.8, 39.3, 30.4, 29.5.

##### Compound **13**

Yellow solid, yield 47%.

ESI-MS for C_27_H_30_N_2_O_4_ (*m*/*z*): [M + H]^+^ 447, [M + Na]^+^ 469, [2M + H]^+^ 893, [2M + Na]^+^ 915.

^1^H NMR (500 MHz, CDCl_3_) δ 7.83 (s, 1H), 7.37 (d, *J* = 11.0 Hz, 1H), 7.31–7.27 (m, 1H), 7.26–7.19 (m, 4H), 7.18–7.13 (m, 1H), 6.55–6.49 (m, 2H), 3.90 (s, 3H), 3.88 (s, 3H), 3.78–3.72 (m, 1H), 3.50 (s, 3H), 3.49–3.46 (m, 1H), 3.42–3.40 (m, 1H), 3.07 (d, *J* = 5.4 Hz, 3H), 2.45–2.28 (m, 2H), 2.26–2.15 (m, 1H), 1.77–1.71 (m, 1H).

^13^C NMR (126 MHz, CDCl_3_) δ 175.8, 155.1, 152.6, 150.6, 150.5, 141.1, 140.0, 138.7, 135.4, 130.5, 128.3, 128.2, 127.0, 126.9, 124.5, 107.2, 107.0, 61.3, 60.6, 60.4, 56.0, 51.9, 39.4, 30.5, 29.5.

##### Compound **14**

Yellow solid, yield 33%.

ESI-MS for C_27_H_29_ClN_2_O_4_ (*m*/*z*): [M + H]^+^ 481/483, [M + Na]^+^ 503, [2M + H]^+^ 961, [2M + Na]^+^ 983/985.

^1^H NMR (500 MHz, CDCl_3_) δ 7.87 (s, 1H), 7.37 (d, *J* = 11.0 Hz, 1H), 7.33 (dd, *J* = 7.2, 1.6 Hz, 1H), 7.30–7.23 (m, 2H), 7.16–7.08 (m, 2H), 6.53–6.49 (m, 2H), 3.90 (s, 3H), 3.88 (s, 3H), 3.81–3.75 (m, 1H), 3.58–3.55 (m, 4H), 3.49–3.44 (m, 1H), 3.07 (d, *J* = 5.4 Hz, 3H), 2.44–2.31 (m, 2H), 2.26–2.16 (m, 1H), 1.81–1.73 (m, 1H).

^13^C NMR (126 MHz, CDCl_3_) δ 175.8, 155.1, 152.6, 150.6, 150.4, 141.1, 138.7, 137.4, 135.4, 133.6, 130.4, 130.2, 129.3, 128.3, 126.8, 124.6, 107.2, 107.0, 61.3, 60.7, 60.3, 56.0, 49.3, 39.3, 30.5, 29.5.

##### Compound **15**

Yellow solid, yield 51%.

ESI-MS for C_27_H_30_N_2_O_5_ (*m*/*z*): [M + H]^+^ 463, [M + Na]^+^ 485, [2M + H]^+^ 925, [2M + Na]^+^ 947, [M − H]^−^ 461, [M + HCOO^−^]^−^ 507.

^1^H NMR (500 MHz, CDCl_3_) δ 7.94 (s, 1H), 7.45 (d, *J* = 11.2 Hz, 1H), 7.40–7.36 (m, 1H), 7.03 (d, *J* = 8.2 Hz, 2H), 6.74 (d, *J* = 8.1 Hz, 2H), 6.62 (d, *J* = 11.4 Hz, 1H), 6.52 (s, 1H), 3.92 (s, 3H), 3.88 (s, 3H), 3.59–3.56 (m, 1H), 3.55 (s, 3H), 3.52–3.46 (m, 1H), 3.27 (d, *J* = 12.0 Hz, 1H), 3.09 (d, *J* = 5.4 Hz, 3H), 2.43–2.36 (m, 1H), 2.35–2.18 (m, 2H), 1.77–1.73 (m, 1H).

^13^C NMR (126 MHz, CDCl_3_) δ 175.1, 156.4, 155.4, 152.8, 151.3, 150.6, 141.1, 139.6, 135.5, 131.4, 130.5 129.5, 126.7, 124.2, 115.6, 108.6, 107.2, 61.4, 61.0, 60.8, 56.1, 51.8, 39.3, 30.5, 29.6.

##### Compound **16**

Yellow solid, yield 56%.

ESI-MS for C_26_H_29_N_3_O_4_ (*m*/*z*): [M + H]^+^ 448, [M + Na]^+^ 470, [2M + Na]^+^ 917.

^1^H NMR (500 MHz, CD_3_CN) δ 8.35 (d, *J* = 5.7 Hz, 2H), 7.66 (s, 1H), 7.36–7.31 (m, 1H), 7.25 (d, *J* = 11.0 Hz, 1H), 7.17 (d, *J* = 5.8 Hz, 2H), 6.61 (s, 1H), 6.55 (d, *J* = 11.2 Hz, 1H), 3.80 (s, 3H), 3.76 (s, 3H), 3.68 (d, *J* = 14.9 Hz, 1H), 3.47 (d, *J* = 14.9 Hz, 1H), 3.39 (s, 3H), 3.35–3.29 (m, 1H), 3.00 (d, *J* = 5.4 Hz, 3H), 2.44–2.36 (m, 1H), 2.26–2.13 (m, 2H), 1.74–1.63 (m, 1H).

^13^C NMR (126 MHz, CD_3_CN) δ 176.2, 156.1, 153.6, 151.3, 150.5, 150.4, 150.2, 141.9, 139.3, 136.2, 130.7, 127.5, 124.6, 123.9, 108.2, 107.7, 61.3, 61.0, 60.9, 56.5, 50.7, 39.5, 30.8, 29.7.

##### Compound **17**

Yellow solid, yield 40%.

ESI-MS for C_25_H_28_N_2_O_4_S (*m*/*z*): [M + H]^+^ 453, [M + Na]^+^ 475, [2M + H]^+^ 905, [2M + Na]^+^ 927.

^1^H NMR (500 MHz, CDCl_3_) δ 7.74 (s, 1H), 7.37 (d, *J* = 11.1 Hz, 1H), 7.29–7.27 (m, 1H), 7.10 (dd, *J* = 5.0, 0.9 Hz, 1H), 6.83–6.80 (m, 1H), 6.76 (d, *J* = 2.9 Hz, 1H), 6.54–6.50 (m, 2H), 3.93 (d, *J* = 14.1 Hz, 1H), 3.90 (s, 3H), 3.88 (s, 3H), 3.65 (d, *J* = 14.1 Hz, 1H), 3.57–3.53 (m, 1H), 3.51 (s, 3H), 3.08 (d, *J* = 5.4 Hz, 3H), 2.45–2.29 (m, 2H), 2.25–2.16 (m, 1H), 1.78–1.71 (m, 1H).

^13^C NMR (126 MHz, CDCl_3_) δ 175.7, 155.1, 152.7, 150.6, 150.2, 143.8, 141.1, 138.8, 135.4, 130.5, 126.8, 126.5, 124.7, 124.4, 124.3, 107.3, 107.0, 61.3, 60.7, 59.9, 56.0, 46.3, 39.4, 30.5, 29.5.

#### 3.3.3. General Procedure for the Synthesis of Colchicine Derivatives **4** and **11**

Compound **4** and **11** were obtained in a two-step procedure. The first step involved the synthesis of aldehyde and the second was the reaction of the obtained aldehyde with the starting compound **3**.

To a solution of appropriate alcohol (10.0 eqiuv.) in DCM was added PCC (30.0 equiv.) and the reaction mixture was stirred at RT for 4 h. After this time the mixture was filtered through thin pad of silica gel and the filtrate containing aldehyde was used in the next step (without further purification). To a solution of compound **3** (1.0 equiv.) in MeOH, one-second of the previously prepared portion of aldehyde was added followed in 15 min by NaBH_3_CN (1.2 equiv.). After two hours, a further portion of the prepared aldehyde and NaBH_3_CN (1.2 equiv.) were added. Reaction progress was monitored by LC-MS. Then the mixture was evaporated under reduced pressure, diluted with EtOAc, washed with 5% NaHCO_3_, brine and dried over Na_2_SO_4_. The residue was purified using column flash chromatography (silica gel; DCM/MeOH, 30/1 *v*/*v*) and next lyophilized from dioxane to give respective compounds.

##### Compound **4**

Yellow solid, yield 13%.

ESI-MS for C_22_H_28_N_2_O_4_ (*m*/*z*): [M + H]^+^ 385, [M + Na]^+^ 407, [2M + Na]^+^ 791.

^1^H NMR (500 MHz, CDCl_3_) δ 7.77 (s, 1H), 7.43 (d, *J* = 11.1 Hz, 1H), 7.26–7.24 (m, 1H), 6.64–6.53 (m, 2H), 3.93 (s, 3H), 3.91 (d, *J* = 4.6 Hz, 3H), 3.90–3.84 (m, 1H), 3.60 (s, 3H), 3.12 (d, *J* = 5.3 Hz, 3H), 2.86–2.69 (m, 1H), 2.59–2.30 (m, 4H), 1.87 (s, 1H), 1.13 (s, 3H).

##### Compound **11**

Yellow solid, yield 43%.

ESI-MS for C_23_H_29_N_2_O_4_ (*m*/*z*): [M + H]^+^ 433/435, [M + Na]^+^ 455/457, [2M + Na]^+^ 887/889.

^1^H NMR (500 MHz, CD_2_Cl_2_) δ 7.59 (s, 1H), 7.33 (d, *J* = 11.0 Hz, 1H), 7.25–7.22 (m, 1H), 6.59–6.47 (m, 2H), 3.86 (d, *J* = 1.9 Hz, 6H), 3.62–3.55 (m, 2H), 3.54 (s, 3H), 3.38–3.36 (m, 1H), 3.06 (d, *J* = 5.5 Hz, 3H), 2.63–2.60 (m, 1H), 2.50–2.39 (m, 2H), 2.31–2.29 (m, 1H), 2.26–2.16 (m, 1H), 1.87–1.82 (m, 2H), 1.74–1.65 (m, 1H).

^13^C NMR (126 MHz, CD_2_Cl_2_) δ 176.0, 155.5, 153.3, 151.1, 151.0, 141.7, 138.9, 135.9, 130.7, 127.3, 124.6, 107.6, 107.4, 61.5, 61.5, 60.9, 56.5, 45.4, 43.8, 39.8, 33.6, 30.8, 29.9.

#### 3.3.4. Synthesis of **10**

Compound **10** was obtained directly from compound **3**. To a solution of compound **3** (1.0 equiv.) in MeOH, acrylonitrile (5.0 equiv.) was added. Reaction progress was monitored by LC-MS. Then the reaction mixture was diluted with EtOAc, washed with 5% NaHCO_3_, brine and dried over Na_2_SO_4_. The residue was purified using column flash chromatography (silica gel; DCM/MeOH) and next lyophilized from dioxane to give the pure product **10** as a yellow solid with a yield of 19%.

ESI-MS for C_23_H_27_N_3_O_4_ (*m*/*z*): [M + H]^+^ 410, [M + Na]^+^ 432, [2M + Na]^+^ 841, [M − H]^−^ 408, [M + HCOO^−^]^−^ 454.

^1^H NMR (500 MHz, CDCl_3_) δ 7.63 (s, 1H), 7.39 (d, *J* = 11.1 Hz, 1H), 7.3–7.27 (m, 1H), 6.60–6.49 (m, 2H), 3.92 (s, 3H), 3.91 (s, 3H), 3.59 (s, 3H), 3.51–3.40 (m, 1H), 3.09 (d, *J* = 5.4 Hz, 3H), 2.84–2.80 (m, 1H), 2.68–2.57 (m, 1H), 2.49–2.40 (m, 3H), 2.36–2.30 (m, 1H), 2.28–2.21 (m, 1H), 1.77–1.71 (m, 1H).

### 3.4. In Vitro Antiproliferative Activity

Cytotoxic activity was tested according to the previously published procedure [20] and detailed information can be found in Appendix A.

### 3.5. Molecular Docking Studies

Molecular docking studies was performed according to the previously published procedure [20] and detailed information can be found in Appendix A.

## 4. Conclusions

In summary, a series of novel double-modified colchicine analogues derivatized at their C7 (single carbon-nitrogen bond) and C10 (methylamine substituent) positions, were devised. All compounds were successfully synthesized, purified and their structures were confirmed by spectroscopic analyses. All target compounds were screened for their in vitro cytotoxicity against A549, MCF-7, LoVo, and LoVo/DX cells to evaluate their anticancer potency.

All presented derivatives were shown to be active at nanomolar concentrations and exhibited better antiproliferative activity than the commonly used anticancer agents doxorubicin and cisplatin. The introduction of methylamino group at position C10 significantly increased the cytotoxicity of a relevant derivative in comparison to that of unmodified colchicine **1**. In addition, the majority of the investigated double-modified derivatives exhibited excellent potency (IC_50_ ≤ 10 nM) against A549, MCF-7, LoVo cell lines. The most toxic to all tumor cells tested was derivative **14** with 2-chlorobenzyl moiety at C7 carbon (IC_50_ = 0.1–1.6 nM). Thus, preliminary conclusions revealed that the choice of the type of chemical bond or substituent at C7 and C10 positions was critical for the cytotoxicity of the compounds.

The calculated values of resistance index (RI) indicated that colchicine-based compounds can be effective against drug-resistant cancer cells. Eleven new derivatives (**3**–**8**, **11**–**13**, **16**–**17**) showed RI ≤ 10, which means that LoVo/DX cells are sensitive to these compounds. Although these compounds have low RI values, they have proved to be non-selective for cancer cells in relation to normal cells (low SI values). Further evaluation should help to find more detailed structure–activity relationships for the drugs overcoming the resistance of tumor cells, which can help in rational drug design efforts in future. In terms of selectivity for cancerous versus normal cells, only one derivative **14** was superior to parent amides **1** and **2**. Replacement of acetyl at C7 carbon with a 2-chlorobenzyl moiety and simultaneous substitution of methoxy group to methylamino group at position C10 allowed obtaining a compound selective towards all four cancer cell lines tested (high SI values from 2.8 to 93.7).

Molecular docking studies showed that apart from the initial amides **1** and **2**, compound **14**, which had the best antiproliferative activity, stood out also in terms of its predicted binding energy and probably binds best into the active site of βI-tubulin isotype. Otherwise, no correlation between in vitro and *in silico* results was observed indicating that other factors than affinity to the target determine their biological activity. These factors may include solubility, membrane permeability as well as off-target interactions including the affinity to P-glycoprotein.

Derivative **14** with strong in vitro antiproliferative activity and high SI values, represents the most promising lead for further development. Test results for this compound are noteworthy and suggest that extended and more detail in vivo research of **14** and similar derivatives is necessary to determine their therapeutic potential as anticancer drugs.

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
