# Peer review of "New Series of Double-Modified Colchicine Derivatives: Synthesis, Cytotoxic Effect and Molecular Docking"

_molecules, 2020, doi:10.3390/molecules25153540_

Round 1

Reviewer 1 Report

There have been numerous previous studies modifying colchicine, given the promising anticancer/tubulin-binding properties of this molecule that are unfortunately compromised by drug resistance and toxicity. This latest paper proposes a double modification at C7 and C10. Straightforward synthetic modification leads to a new series, from which molecules with potent activity within standard 2D cell lines emerges. Compound 14 stands out here, including some selectivity towards the drug resistant cell line LoVo/DX. Selectivity for cancer cell lines over normal cell line (BALB/3T3) is moderate for most compounds. The authors should be careful not to read too much significance into these results on cell lines that poorly represent the patient's tumour in vivo. However the results will be of interest to researchers within the field.

Further rationalisation for the antiproliferative activity is enabled by studying the binding to target protein structure using computational modelling. Again, the authors should be careful not to overstate the significance of this data, and I recommend that the bulk of the docking results for each analogue (Table 2) are moved to Supplementary Information, except for the most significant analogues.

Although the work and write-up is carried out to a high standard, the significance of the results are only moderate. This is because the paper does not directly address the significant drawbacks of colchicine - P-gp efflux and neurotoxicity. If these points can be addressed (e.g. using in silico predictive methods, or in vitro experiments), this would make a more compelling and significant paper. However, the manuscript is just satisfactory as it stands, once the authors have given a statement of the limitations of the cell line and molecular modelling data, and some of the detail of Table 2 is moved to Supplementary.

Author Response

Ad both Reviewers:

We thank the Reviewers for their thorough evaluation of our study and their insightful comments which helped us strengthen our article. We hope that the Reviewers will find our corrections and explanations satisfactory.

Ad Reviewer 1:

  1. Referring to the comment: “I recommend that the bulk of the docking results for each analogue (Table 2) are moved to Supplementary Information, except for the most significant analogues.”, the Table 2 on page 8 was reformatted according to the recommendations. Table 2 shows the results for the starting compounds 1-3 and the most cytotoxic compound 14. Data for the remaining analogues are shown in Supplementary Materials in Table S1.
  2. In response to the comment: “(...)the manuscript is just satisfactory as it stands, once the authors have given a statement of the limitations of the cell line and molecular modelling data”, there is information in the main text about the need for further research and limitations (we have also added some new sentences):

Page 5, lines 147-150: “Additionally, it should be mentioned that the antiproliferative activity depends on the type of the tested cell lines and is different for different cells. In vitro tests allow the initial determination of biological activity, but in vivo studies are necessary to determine the therapeutic potential”.

Page 5, lines 159-151: N-(2-chlorobenzyl)-10-methylaminocolchicine 14 therefore emerges as a very promising molecule (the lowest IC50 and the highest SI) for further more detail research (in vitro but also in vivo) as a potential anticancer agent”.

Section 3. In Silico Determination of the Molecular Mode of Action provides information on the limitations of computer modelling studies: solubility, membrane permeability, affinity to P-glycoproteins, other efflux pumps, other tubulin isotypes etc. (pages 7-8, lines 217-235).

Page 14, lines 445-447: “Further evaluation should help to find more detailed structure-activity relationships for the drugs overcoming the resistance of tumour cells, which can help in rational drug design efforts in future”.

  • Page 14, lines 454-457: “(...)indicating that other factors than affinity to the target determine their biological activity. These factors may include solubility, membrane permeability as well as off-target interactions including the affinity to P-glycoprotein”.
  • Page 14, lines 459-461: “Test results for this compound are noteworthy and suggest that extended and more detail in vivo research of 14 and similar derivatives is necessary to determine their therapeutic potential as anticancer drugs”.

Reviewer 2 Report

In this article Authors reported synthesis, biological evaluation and docking studies of 17 compounds strictly related to colchicine. The paper is well written, well organized and interesting from the point of view of biological results. In my opinion it can therefore be published with minor revisions.

In detail:

  • The most promising compound 14 presents a chlorine atom in orto position. No isoster compounds (para and meta isosters) are reported; authors could explain why these compounds are not synthesized (some phrases in introduction or in conclusion).
  • If it is possible interpretation of 1H NMR should be added.
  • Page 15, lines 369-379: this experimental procedure is not clear; please specify yields of aldehyde synthesis and percentage pf DCM/MeOH used for flash chromatography prurification.

Author Response

Ad both Reviewers:

We thank the Reviewers for their thorough evaluation of our study and their insightful comments which helped us strengthen our article. We hope that the Reviewers will find our corrections and explanations satisfactory.

Ad Reviewer 2:

  1. Referring to the comment: “The most promising compound 14 presents a chlorine atom in orto position. No isoster compounds (para and meta isosters) are reported; authors could explain why these compounds are not synthesized (some phrases in introduction or in conclusion).” we have added explanation addressing this issue in the main text on page 2 (lines 67-69).

It is: “The derivatives presented in this work are double-modified colchicine derivatives with the amine groups at C7 and C10 positions. On the basis of the obtained preliminary structure-activity relationship (SAR) studies, we will be able to design new promising compounds for further more extended research.”

Additionally, the section 4. Conclusions contain information about the need to test derivatives similar to compound 14 (page 14, lines 459-462 “(...)Test results for this compound are noteworthy and suggest that extended and more detail  in vivo research of 14 and similar derivatives is necessary to determine their therapeutic potential as anticancer drugs.”).

  1. Comment: “If it is possible, interpretation of 1H NMR should be added”.

Response: For a discussion of the results of the NMR analysis, see section 2.1. Chemistry, we have added some new more detail information (page 3, lines 100-109). It is: “The signals corresponding to -OCH3 group at position C10 of 1 in the 1H NMR and 13C NMR spectra were observed as singlets at 4.0 ppm and at 56.5 ppm, respectively. After replacement of this group in the tropolone ring of colchicine by -NHCH3 group in compounds 2-17, new signals appeared, at 3.0-3.1 ppm as a doublet and 7.2-7.3 ppm as a quartet in 1H NMR spectra and at 29.5-29.9 ppm in 13C NMR spectra. The signals characteristic of the acetyl group in colchicine 1 appeared at 1.9 ppm (-CH3) and 8.6 ppm (-NH-) in 1H NMR and at 22.7 ppm (-CH3) and 170.3 ppm (-C=O) in 13C NMR, respectively. These signals are no longer observed in the NMR spectra of 3-17 derivatives. The signal assigned to the hydrogen atom at C7 has been shifted from 4.6 ppm for unmodified colchicine 1 to 3.4-3.6 ppm for compounds 3-17.”

  1. In response to the Reviewer’s comment: “Page 15, lines 369-379: this experimental procedure is not clear; please specify the yields of aldehyde synthesis and percentage of  DCM/MeOH used for flash chromatography purification.”, we have reformatted the synthesis description and added some information about the composition of solvents used for flash chromatography purification on page 12 (lines 379-390). We have not given  the yield of  the synthesis of the aldehyde as it has not been determined. The aldehyde was used in the reductive alkylation without purification.